# ©Plug-in Authorization for Human Copyright Protection in Text-to-Image Model

**Chao Zhou**[*]                                                                    *zc1696190340@mail.ustc.edu.cn*
*University of Science and Technology of China*

**Huishuai Zhang**[*]                                                                 *zhanghuishuai@pku.edu.cn*
*Wangxuan Institute of Computer Technology, Peking University*
*National Key Laboratory of General Artificial Intelligence*

**Jiang Bian**                                                                          *jiabia@microsoft.com*
*Microsoft Research*

**Weiming Zhang**                                                                   *zhangwm@ustc.edu.cn*
*University of Science and Technology of China*

**Nenghai Yu**                                                                          *ynh@ustc.edu.cn*
*University of Science and Technology of China*

**Reviewed on OpenReview:** *https://openreview.net/forum?id=4o8lIFkpn2*

## Abstract

This paper addresses the contentious issue of copyright infringement in images generated by text-to-image models, sparking debates among AI developers, content creators, and legal entities. State-of-the-art models create high-quality content without crediting original creators, causing concern in the artistic community and model providers. To mitigate this, we propose the ©Plug-in Authorization framework, introducing three operations: addition, extraction, and combination. Addition involves training a ©plug-in for specific copyright, facilitating proper credit attribution. The extraction allows creators to reclaim copyright from infringing models, and the combination enables users to merge different ©plug-ins. These operations act as permits, incentivizing fair use and providing flexibility in authorization. We present innovative approaches, "Reverse LoRA" for extraction and "EasyMerge" for seamless combination. Experiments in artist-style replication and cartoon IP recreation demonstrate ©plug-ins' effectiveness, offering a valuable solution for human copyright protection in the age of generative AIs. The code is available at `https://github.com/zc1023/-Plug-in-Authorization.git`

## 1 Introduction

Large foundation models Brown et al.; Touvron et al. (2023); Radford et al. (2021); OpenAI (2023); Rombach et al. (2022b) are trained with extensive, high-quality datasets like The Pile Gao et al. (2020), C4 Raffel et al. (2020), LAION Schuhmann et al. (2022) and other enormous undisclosed data sources, which definitely contain copyrighted human contents. At the same time, these models not only excel at generating content based on user prompts cha; OpenAI (2023); Rombach et al. (2022b); Ramesh et al. (2021; 2022), but also have the potential of memorizing the exact training data thanks to the huge capacity in their gigantic numbers of parameters Carlini et al. (2021; 2023a).

Such training procedure and utilization of AI models have sparked copyright infringement concerns among content providers, artists, and users. A notable instance is the lawsuit filed by The New York Times against

---

[*]indicates equal contribution. Corresponding to Huishuai Zhang.

OpenAI and Microsoft NYT, alleging the unauthorized use of a vast number of articles for the purpose of training automated chatbots. The lawsuit seeks the destruction of the allegedly infringing chatbots and their associated training data. Similar concerns and legal actions are also emerging in the field of text-to-image generation law.

Indeed, these concerns are well justified as these powerful models could disrupt the existing reward system in creative arts, adding anxiety to the content providers and artist community. The proficiency of AI in generating artworks that rival human creations is noteworthy, particularly in its ability to replicate characters from major intellectual properties (IPs). For instance, the use of stable diffusion models Rombach et al. (2022a), combined with controlled generation techniques like ControlNet Zhang et al. (2023b), enables users to effortlessly create well-known characters, such as those from Disney. This ease of replication significantly lowers the barriers to potential copyright infringement, raising concerns about increased piracy risks.

One debating point is whether using copyrighted material to train machine learning models is prohibited by copyright laws. It is known that copyright does not ban all forms of copying or replication due to the *fair use* doctrine, which allows certain copying and distribution if it can be justified as fair use. It is not clear whether AI companies can successfully argue that their training procedures fall under this 'fair use' exception in copyright laws NYT; law. Additionally, academic research is underway to develop methods ensuring AI models do not generate copyrighted concepts, as seen in works like Vyas et al. (2023).

In this paper, we step back and advocate to rethink the motivation for enforcing copyright laws. Copyright is a type of intellectual property that intends to protect the original expression of ideas in creative works, which can include literary, artistic, or musical forms cop. The foundational goal of copyright laws, as stated in Article I, Section 8, Clause 8 of the U.S. Constitution USC, is "To promote the Progress of Science and useful Arts, by securing for limited Times to Authors and Inventors the exclusive Right to their respective Writings and Discoveries". The primary objective of copyright is to incentivize authors to create new works and to facilitate the dissemination of these works to the public by granting them property rights. However, existing generative AI models present challenges in appropriately attributing proper rewards to the copyright holders, which can significantly impact society. Artists, who depend on attribution for recognition and income, may be affected. Additionally, domain experts contributing to knowledge exchange websites like StackOverflow and Quora might hesitate to provide answers if they do not receive reasonable rewards. This situation could backfire on machine learning, as generative models might soon face a shortage of fresh data due to reduced contributions from these sources.

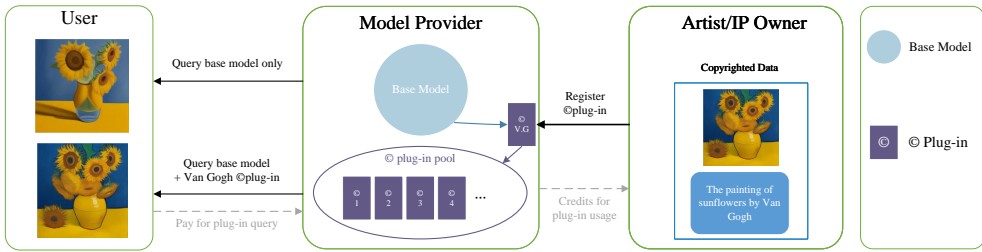

Figure 1: ©**Plug-in Authorization Process.** The authorization process consists of three types of entities: user, model provider, and IP owners (artists). Users can generate copyrighted images only by accessing the relevant plug-in. The model provider offers services to users, tracks usage of plug-ins, and attributes rewards to the IP owner. The IP owners can achieve authorization by registering their ©plug-ins through *addition* or *extraction*. These ©plug-ins form a pool where users can get ©plug-ins to produce content with the IP owner's authorization.

To address the attribution challenge in generative AI models, the concept of *Stable Attribution* Troynikov (2023) has been proposed, aiming to credit artists and share revenue with creators based on their contributions. Specifically, Stable Attribution attempts to trace back an AI-generated image to the most similar examples in the training dataset. However, achieving this with reasonable cost and ensuring fairness is challenging, given the vast size and heterogeneous nature of the training set. Content providers and model owners may have completely different views on the evaluation of the content NYT. While copyright data is unique, its impact

Figure 2: **Three foundational operations achieving ©Plug-in Authorization: *addition*, *extraction*, and *combination*.** The plug-in can be created by addition if the copyrighted work is new to the base model. Meanwhile, the plug-in can be created by extracting from the base model if the copyrighted work is already infringed by the base model. Once a pool of ©plug-ins is constructed, the *combination* operation can merge multiple ©plug-ins featuring the generation of multiple concepts and leave a non-infringing model excluding all multiple concepts.

on model performance may not be able to be fairly measured, especially when compared to the benefits derived from the vast size of open-domain public data. The value of artistic works might be underestimated by model owners. Addressing these issues is essential to create a balanced ecosystem in generative AI.

In addressing the challenge of practical attribution in generative models, we introduce the "Copyright Plug-in Authorization" framework (see Figure 1), designed to align with existing Intellectual Property (IP) management practices. This framework involves base model providers, like Stability AI, functioning as repositories for copyright plug-ins. Copyright holders, such as artists, can register their works as plug-ins, receiving rewards for their use. End users, in turn, pay for the generation of images involving copyrighted concepts using these plug-ins. This system offers positive incentives for all involved: copyright holders are compensated for their creative contributions, end users can use copyrighted plug-ins without risking infringement, and base model providers profit from plug-in registration and model usage. The framework also facilitates explicit tracking of copyrighted work usage, ensuring a fair and straightforward reward system. By successfully implementing this authorization process, we can enable a more equitable distribution of copyright benefits across the generative model landscape.

Technically to enable an effective and efficient copyright authorization, the plug-ins, as permits, should be easily created by *addition* if copyrighted works are new to the base models, or by *extraction* if the copyrighted works are already infringed by the base model. Moreover, the plug-ins should be easily *combined*, which allows copyright holders to merge multiple plug-ins into a new one or enables end users to generate images with multiple copyrighted works. Meanwhile, for efficient execution, these operations should be implemented as light adaptations to the base model, e.g., parameter-efficient tuning methods or prompt designs.

In this paper, we introduce three foundational operations - *addition, extraction*, and *combination* - implemented using the Low-Rank Adaptor (LoRA) method Hu et al. (2022). These operations are essential for realizing the Copyright Plug-in Authorization (Figure 2 for an overview).

It is noteworthy that Civitai (civ) represents a commendable attempt to instantiate the *addition* operation, as users can train and share LoRA components to generate corresponding figures. However, the operations *extraction* and *combination* are currently not publicly available and pose greater challenges.

The *extraction* operation involves separating the generative model into a non-infringing base model and some copyrighted plug-ins. A conventional approach might involve retraining the model from scratch using only non-infringing data, and then applying LoRA with copyrighted data. However, this method is impractical, if not impossible, due to high training costs and complex data-cleaning processes. Alternatively, this paper introduces a "Reverse LoRA" approach to extract a plug-in from an infringing base model. This process begins by capturing the target concept: we LoRA-tune the model on the target concept and then take the negative of the LoRA weights to achieve concept destruction. Then we fine-tune the LoRA on surrounding contexts to repair the non-infringing model's contextual generation ability. Finally, we reverse the LoRA to be the ©plug-in.

The *combination* operation entails merging multiple copyrighted plug-ins into a unified one. Simply adding these plug-ins together could lead to unpredictable results due to the correlation among copyrighted plug-ins. In this paper, we have successfully developed a method to fuse multiple components. We introduce "EasyMerge", a method termed "data-free layer-wise distillation" for the combination process. Inspired by conditional generation in generative models, we utilize a LoRA component designed to learn the layer-wise outputs of ©plug-ins under corresponding conditions. Consequently, the LoRA component can mimic the behavior of these ©plug-ins when subjected to the corresponding conditions, effectively achieving the combination of ©plug-ins.

Our contributions are summarized as follows:

- **Conceptual contribution: A ©Plug-in Authorization framework.** We advocate to solve the problem of copyright infringement in foundation models with a ©Plug-in Authorization framework. It can offer a fair and practical solution for the attribution challenge in text-to-image generative models. We further introduce three operations *addition, extraction* and *combination* to instantiate the framework with efficient human content copyright authorizations.

- **Technical contribution: A novel "Reverse LoRA" algorithm for *extraction*.** It can effectively *extract* copyrighted concepts from the base model, achieving competitive performance for concept extraction with flexible plug-ins.

- **Technical contribution: A novel "EasyMerge" approach for *combination*.** It is a data-free layer-wise distillation approach, which can effectively and efficiently address the challenge of combining multiple LoRA components.

The structure of the paper is as follows. Section 2 introduces the "©Plug-in Authorization" framework and delves into the three operations *addition, extraction* and *combination*. Section 3 presents experiments to validate the effectiveness of the proposed operations. Section 5 concludes the paper, offering a discussion on the limitations of our work.

## 2 ©Plug-in Authorization with Addition, Extraction and Combination

As detailed in the Introduction, we implement the "©Plug-in Authorization" by utilizing the publicly available pretrained diffusion generative model, Stable Diffusion (Rombach et al., 2022a), along with LoRA components (Hu et al., 2022). It is important to note that our framework is not confined to specific model structures, thereby facilitating compatibility with other foundation models such as the GPT series (Brown et al.). Additionally, it is capable of synergizing with various light fine-tuning or prompt tuning techniques (Li & Liang, 2021; Lester et al., 2021; Edalati et al., 2022; Hyeon-Woo et al., 2021). In the subsequent sections, we revisit the fundamentals of diffusion generative models and introduce the three basic operations of the framework, along with our innovative algorithms.

### 2.1 Preliminary on Diffusion Generative Model

Diffusion models (Sohl-Dickstein et al., 2015; Song et al., 2020; Ho et al., 2020) are probabilistic models designed to learn a data distribution. In the forward pass, Gaussian noises are successively added $T$ times to an image $X_0$, thereby creating a sequence $X_0, ..., X_T$ that simulates a Markov process. Conversely, the reverse process trains the model to denoise, effectively emulating the reversal of the Markov Chain. New images are generated by initially sampling random Gaussian noises and then denoising them using the model. Importantly, this process can be conditioned on inputs, such as a prompt text $c$. The denoising process, denoted as $\Phi_{(w)}(X_t, c, t)$, is trained to predict the noise under the textual prompt $c$ at any timestep $t \in [0, T]$, as outlined in (1).

$$\arg\min_w \ \mathbb{E}_{\epsilon, X, c, t} \|\Phi_{(w)}(X_t, c, t) - \epsilon\|^2 \tag{1}$$

Recent advancements have introduced *latent diffusion models* as a solution to mitigate the drawbacks associated with evaluating and optimizing models in pixel space, such as low inference speed and high training

costs. These latent diffusion models operate within a compressed latent space, exemplified by publicly available models such as the Stable Diffusion Model (SDM), as detailed in Rombach et al. (2022a). The SDM architecture consists of a variational autoencoder (VAE) that maps images to latent space, a U-Net that learns the diffusion process, and a CLIP encoder for text embedding. Our work primarily focuses on the attention structure within the U-Net, which has been identified as the most influential component in diffusion models.

To implement the "©Plug-in Authorization", we incorporate three foundational operations into the Stable Diffusion Model (SDM): *addition*, which allows copyright owners to add a plug-in for their works; *extraction*, which enables owners to extract a plug-in from an infringing base model; and *combination*, which permits users to merge plug-ins for multiple copyrighted concepts. The *addition* operation employs LoRA components that are added to SDM's attention matrices, and these are then trained with copyrighted data. While the specifics of the *addition* operation are covered on existing model-sharing platforms like civ, our discussion will primarily focus on the *extraction* and *combination* operations in the subsequent sections.

## 2.2 Extraction: Reverse LoRA

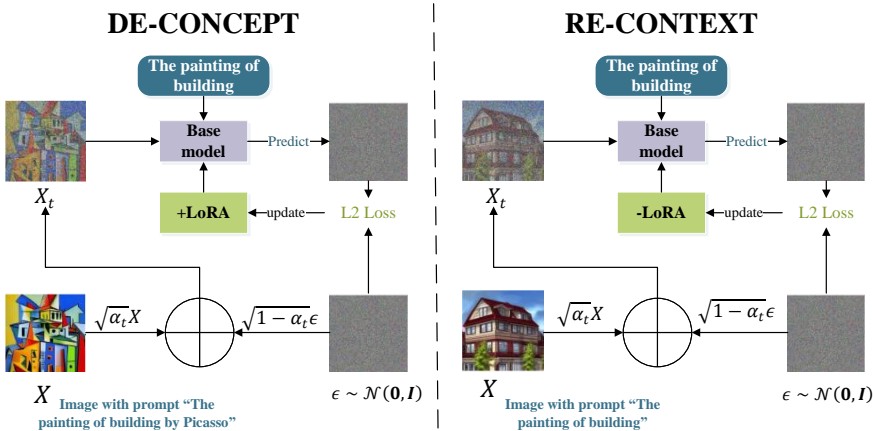

Figure 3: **The method of *extraction*** consists of two steps:*de-concept* and *re-context*. The de-concept step tries to capture the target concept "Picasso" by tuning the LoRA component to match copyrighted images with the contextual prompt "The painting of building". In the re-context step, we reverse the LoRA (so that successfully forget "Picasso") and then further tune the LoRA with surrounding contextual prompt and non-copyrighted image pairs, to ensure the capabilities of contextual generation.

In our endeavor to achieve *extraction*, we introduce a method known as "Reverse LoRA". This approach encompasses two crucial steps to effectively extract the target copyrighted concept while maintaining the ability to generate contextual concepts. The initial step, termed the *de-concept* step, involves removing the target concept from the base model. The subsequent step is a counterbalance to the *de-concept* step, involving relearning the surrounding semantic context, referred to as the *re-context* step. Figure 3 illustrates these two steps in the context of extracting the target concept "Picasso".

### 2.2.1 Step1: De-concept

Our goal is to extract Picasso-related information from the base model to a copyright plug-in using LoRA ($w_L$). This involves identifying the information that represents "Picasso" via an alignment process and documenting the changes in model parameters that occur during the alignment process.

Specifically, we align copyrighted image generation, e.g., images of "the painting of a building by Picasso", with the prompt without copyrighted text, e.g., "the painting of a building" in the base infringing model. The alignment objective is mathematically expressed as follows:

$$\mathbb{E}[\Phi_{(w)}(\epsilon, c^*, t)] = \mathbb{E}[\Phi_{(w+w_L)}(\epsilon, c, t)] \tag{2}$$

where $\Phi$ is the denoising function, $w$ denotes the original network parameter, $w_L$ is the LoRA component, $c$ is the prompt "the painting of a building",$c^*$ is the prompt "the painting of a building by Picasso", $\epsilon$ is the initial noise, and $t$ is the sampling timestep.

To achieve the alignment as defined in (2), we optimize the following objective function with respect to the LoRA parameters $w_L$, while keeping all other model parameters frozen. The objective function is defined as follows:

$$\arg\min_{w_L} \ \mathbb{E}_{\epsilon, X^*, c, t} \|\Phi_{(w+w_L)}(X_t^*, c, t) - \epsilon\|^2 \tag{3}$$

where $X^*$ is the copyrighted image (or generated by the infringing model with the prompt "the painting of building by Picasso"), $X_t^* = \sqrt{\alpha_t}X^* + \sqrt{1 - \alpha_t}\epsilon$ is the noisy version of $X^*$, $c$ is the prompt of "the painting of building", $w$ is the original network and $w_L$ is the LoRA weight. This optimization aims to adjust $w_L$ so that it effectively captures the desired information related to the target concept.

By incorporating such a LoRA component, the base model can generate Picasso-style images even when the prompts do not explicitly mention "Picasso". Hence, the LoRA represents the copyrighted Picasso style, and $w - w_L$ would give us a non-infringing model, which can thought of as an analogy of a 'negative LoRA'. However, directly using $w - w_L$ as the non-infringing model compromises its ability to generate images with surrounding context, e.g., "the painting of a building", as shown in Figure 7 in Appendix A.1. This observation leads us to further tune the LoRA with pairs of images and texts of surrounding semantic context.

### 2.2.2 Step2: Re-context

To mitigate the performance degradation of the non-infringing model when generating images with contextually related prompts, we introduce a re-context step following the de-concept step. This step involves fine-tuning the LoRA component with images and textual prompts of surrounding contexts, e.g., "the painting of a building". To curate the dataset, we randomly generate images with the base model using the contextual prompt "the painting of a building", while leveraging the negative prompt (Ho & Salimans, 2022) "Picasso" to steer the generation as far away from the target concept "Picasso" as possible.

Specifically, we further optimize $w_L$ with the objective,

$$\arg\min_{w_L} \ \mathbb{E}_{\epsilon, X, c, t} \|\Phi_{(w-w_L)}(X_t, c, t) - \epsilon\|^2, \tag{4}$$

where $X, c$ represent the constructed pairs (image, prompt) to recover the generation capability of surrounding contexts.

Overall, after the de-concept step, the model $w - w_L$ is unable to generate images in the Picasso style, yet it performs well with surrounding prompts thanks to the re-context step. Therefore, through the *extraction* operation, we obtain a non-infringing model $\tilde{w} = w - w_L$ and a ©plug-in $w_L$. By incorporating the ©plug-in, the model is restored to the original base model $w$, regaining the capability of successfully generating the artworks in the "Picasso" style. The intermediate results of the *extraction* process are visually illustrated in Figure 7 in Appendix A.1, showcasing the successful extraction of the targeted copyright, while preserving the model's ability to generate images with surrounding contexts.

### 2.3 Combination: EasyMerge

In this section, we consider the operation *combination*. The combination of existing ©plug-ins becomes essential when aiming to generate an image featuring both "Snoopy" and "Mickey" concepts.

It is worth noting that simply adding these plug-ins together could yield unpredictable outcomes due to inherent correlations among these plug-ins. To facilitate the combination of multiple copyrighted concepts, we propose a novel approach named *EasyMerge*. This method employs a data-free, layer-wise distillation technique that only requires plug-ins and corresponding text prompts. Furthermore, with layer-wise distillation, EasyMerge achieves efficient combination in just a few iterations. The versatility of EasyMerge extends beyond the current context, potentially also applicable in other scenarios like continual learning.

Specifically, we use a new LoRA component $w_L$ to mimic the functionalities of each plug-in that needs to be combined. The objective is defined as follows:

$$\arg\min_{w_L} \sum_{k \in S, j \in S_L} \mathbb{E}_{\epsilon,t} \|\phi_{w-w_L}^j(\epsilon, c_k, t) - \phi_{w-w_{L_k}}^j(\epsilon, c_k, t)\|^2, \tag{5}$$

where $S$ is the set of text prompts to be combined, $S_L$ is the set of layers that are added with LoRA components, and $\phi^j$ is the output of layer $j$'s LoRA component. Similarly to the previous section, $w$ denotes the base model parameter, $w_L$ denotes the combined plug-in, $c_k$ denotes the prompt $k$, $w_{L_k}$ is the plug-in of context $c_k$, $\epsilon$ is initial noise and $t$ is the sampling timestep of the diffusion process. The non-infringing model $w - w_L$ is that simultaneously excludes multiple styles related to $c_k$, which is called the *combination* of *extraction*. Algorithm 1 describes concrete steps of optimizing the objective (5).

---

**Algorithm 1:** Combination: EasyMerge method

---

**Input:** A set $S$ of indices of plug-ins to be combined, base model $w$, diffusion step $T$
**Output:** Combined LoRA $w_L$
**repeat**
    **for** $w_{L_i}, c_i \in S$ **do**
        $t \sim \text{Uniform}([1...T])$;
        $\epsilon \sim \mathcal{N}(0,1)$;
        AddHook($w_{L_i}$) ;                 // Capture input $I_{w_{L_i}}^j$ and output $O_{w_{L_i}}^j$ for each layer $j$
        $I_{w_{L_i}}^j, O_{w_{L_i}}^j \leftarrow \Phi_{w+w_{L_i}}(\epsilon, c_i, t)$ ;              // Denoise to obtain features
        $O_{w_L}^j \leftarrow \phi_{w_L}^j(I_{w_{L_i}}^j)$ ;              // Get layer-output through new LoRA
        $\mathcal{L} \leftarrow \sum_{j \in S_L} \|O_{w_L}^j - O_{w_{L_i}}^j\|$;
        $w_L \leftarrow w_L - \nabla_{w_L}\mathcal{L}$;
    **end**
**until** *convergence*;

---

# 3 Experiments to Verify Basic Operations

As a position paper, we regard our primary contribution as the proposal of the copyright authorization framework. Nonetheless, we also aim to validate the practical effectiveness and efficiency of the proposed basic operations. Given that the *addition* operation has already been well demonstrated by existing practices, we focus on evaluating the *extraction* and *combination* operations. We choose two typical scenarios of copyright infringement: artist-style replication and cartoon intellectual property (IP) recreation.

## 3.1 Experiment Setup, Metrics and Baselines

**Experiment Setup.** In all experiments, we fine-tune the attention component in the U-Net architecture of Stable Diffusion Model v1.5, as described in Rombach et al. (2022a).

For the *extraction* operation, we need to generate data with pre-trained models. For the case of extracting a given artistic style, we leverage ChatGPT (cha) to generate 10 common content in paintings. In the de-concept step, for each iteration, we select one of these content to generate 8 images with prompts " The painting of [content] by [artist]". Similarly, during re-context step, we select imagery to generate 8 images with prompts " The painting of [content]" while using negative prompts "by [artist]". For the case of extracting a particular IP character, it follows the same procedure as above except that the prompts become "The cartoon of the [IP character]" for the de-concept process and "The cartoon of the [character]" for the re-context process, respectively. For both the de-concept process and the re-context process, the training consists of 10 iterations, with each iteration 30 epochs. We use a learning rate of 1.5e-4, $T = 50$ steps for the diffusion process, and a rank of 40 for LoRA.

For the *combination* operation, we use a learning rate of 1e-3 and a rank value of 140 for LoRA.

**Metric.** To evaluate the effectiveness of the *extraction* operation, we measure the discrepancy between the set of images generated by the base model and that generated by the non-infringing model after extraction with the same set of prompts. We want to observe a large discrepancy when the prompts are with target concepts while having a small discrepancy with surrounding concepts. This means that the *extraction* operation achieves its goal: the non-infringing model cannot generate images with target concepts but can generate high-quality images with surrounding prompts.

We acknowledge that for image generation tasks, the ultimate evaluation criterion is human judgment. Therefore, we provide the generated images from various scenarios for readers' assessment. Nevertheless, to reduce costs and facilitate comparisons with existing approaches, we also employ an objective metric known as the *Kernel Inception Distance* (KID) (Bińkowski et al., 2018) to quantify the aforementioned discrepancy. KID is akin to the *Fréchet Inception Distance* (FID) (Heusel et al., 2017) but is considered to exhibit less bias and possess asymptotical normality. Moreover, we also employ the Learned Perceptual Image Patch Similarity (LPIPS) (Zhang et al., 2018) to quantify the discrepancy of artistic style artworks. LPIPS is a robust measurement tool that effectively captures differences in human perception between two images, offering a comprehensive evaluation of stylistic variations in generated artworks.

**Baseline**. We compare our *extraction* operation with the concept ablation approach (Kumari et al., 2023) and Erased Stable Diffusion (ESD) (Rohit Gandikota and Joanna Materzyńska and Jaden Fiotto-Kaufman and David Bau, 2023), which achieve concept removal by aligning latent representations of target concepts with those of anchor concepts.

In general, we find it hard to compare the results with existing methods because of the complex setups in image generation, e.g., the tuning steps and the trade-off between removing the target concept and keeping the surrounding concept. Therefore we take a conservative approach and only consider the generation with similar scenarios and the same metric as in the original paper.

## 3.2 Extraction and Combination of Artists' Styles

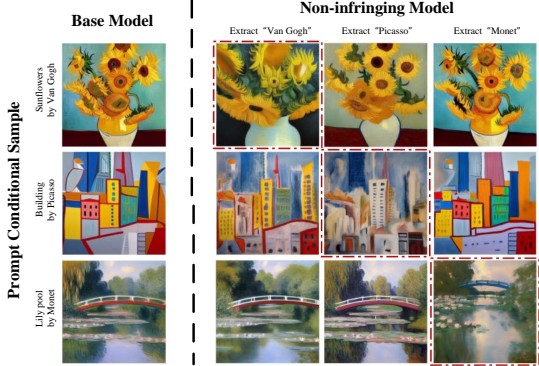
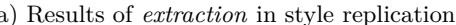

(a) Results of *extraction* in style replication

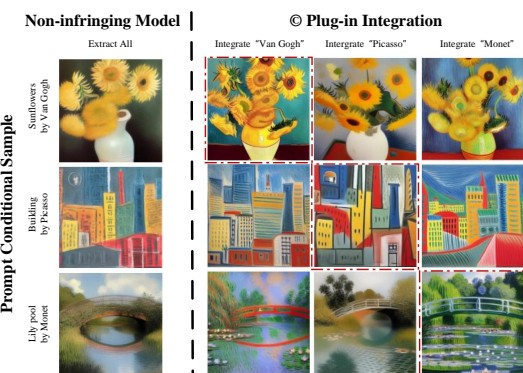

(b) Results of *combination* in style replication

Figure 4: **Results of style replication**. In Figure (a), we show samples from different non-infringing models in each column. Each non-infringing model exhibits a deficiency in one style generation ability, with all other style generation capabilities remaining unaffected. In Figure (b), we present samples generated after integrating certain ©Plug-ins in each column. Each of these ©Plug-ins serves to exclusively restore the generation of one particular style, while the generation of other styles continues to exhibit diminished performance.

**Extraction.** We extract artist styles from the Stable Diffusion V1.5, referred to as the "base model". We consider three renowned artists: (1) Vincent van Gogh, (2) Pablo Ruiz Picasso and (3) Oscar-Claude Monet. The results of individual style extractions are visually presented in Figure 4(a). These images showcase outputs generated by both the base model and the non-infringing model, encompassing both the target styles and surrounding styles.

In Figure 4, images within red boxes represent the target styles, while the rest images embody surrounding styles. A notable contrast is observable between the images within the red boxes and those generated by the base model. However, the images representing surrounding styles exhibit a substantial similarity to those generated by the base model. This demonstrates the success of the *extraction* operation in isolating the target style from the base model while preserving the quality of images with surrounding styles.

Table 1: **Quantitative comparison with baselines in artist-style extraction.** Compared to Concepts-Ablation, ours extracts the target style more thoroughly, and compared to ESD, ours enjoys less damage to surrounding styles.

| Metrics | Methods | Target style ↑ | Surrounding style ↓ |
|---|---|---|---|
| KID$\times 10^3$ | EXTRACTION (OURS) | **187** | 32 |
| | CONCEPTS-ABLATION | 42 | 12 |
| LPIPS | EXTRACTION (OURS) | 0.31 | **0.14** |
| | ESD | 0.38 | 0.21 |

In Table 1, we employ quantitative metrics to evaluate the effectiveness of the *extraction* operation in comparison to baseline methods. Our method demonstrates notable improvements, as indicated by the KID metric increasing from 42 to 187 for the target style when compared to Concepts-Ablation (Kumari et al., 2023). This increase indicates an enhanced removal of the target style. Additionally, in a comparative assessment with the Erasing method (Rohit Gandikota and Joanna Materzyńska and Jaden Fiotto-Kaufman and David Bau, 2023), our method achieves a reduction in LPIPS from 0.21 to 0.14 for surrounding styles. This reduction implies less degradation of the surrounding artistic styles, affirming our method's ability to preserve the quality of generated images when using surrounding style prompts.

**Combination.** In this part, we show the effectiveness of the *combination* operation for extracting multiple artist styles and then adding them back with corresponding plug-ins.

Given three artistic styles of Van Gogh, Picasso, and Monet and their ©plug-ins, we first extract these three styles from the base model, producing a non-infringing model, which is illustrated in the leftmost column of Figure 4(b). Notably, all the images generated by the non-infringing model significantly differ from those generated by the base model in the leftmost column of Figure 4(a). This underscores the efficacy of the combination of multiple *extraction* operations.

We then individually integrate each style copyright plug-in into the non-infringing model. The images highlighted within red boxes represent the target style achieved after integrating the respective copyright plug-in. Notably, the target style images after integration are distinctly different from those produced by the non-infringing model, showing a closer resemblance to the images generated by the base model. This observation indicates that the copyright plug-in can reinstate the model's ability to create artworks in the target styles, without infringing upon the copyright restrictions associated with other artistic styles.

### 3.3 Extraction and Combination of Cartoon IPs

In the context of intellectual property (IP) recreation, we demonstrate the capabilities of our framework through both *extraction* and *combination* operations. Specifically, Figure 5 displays the outcomes of extracting three iconic IP characters: Mickey, R2D2, and Snoopy. The images framed in red boxes were generated by the non-infringing model using prompts specific to the target IP, after extraction. These images significantly deviate from those produced by the base model. In contrast, images outside the red boxes, which represent other IPs, show a resemblance to those generated by the base model, indicating the targeted nature of the extraction process.

Our approach to IP extraction effectively isolates the specified IP, ensuring that the generation capabilities for other IPs remain intact. The efficacy of our extraction method in the realm of IP recreation is quantified in Table 2, where we document a notable improvement, i.e., approximately a 2.6-fold increase in the Kernel Inception Distance (KID) metric for the targeted IP, while the KID metrics for other IPs remain relatively stable.

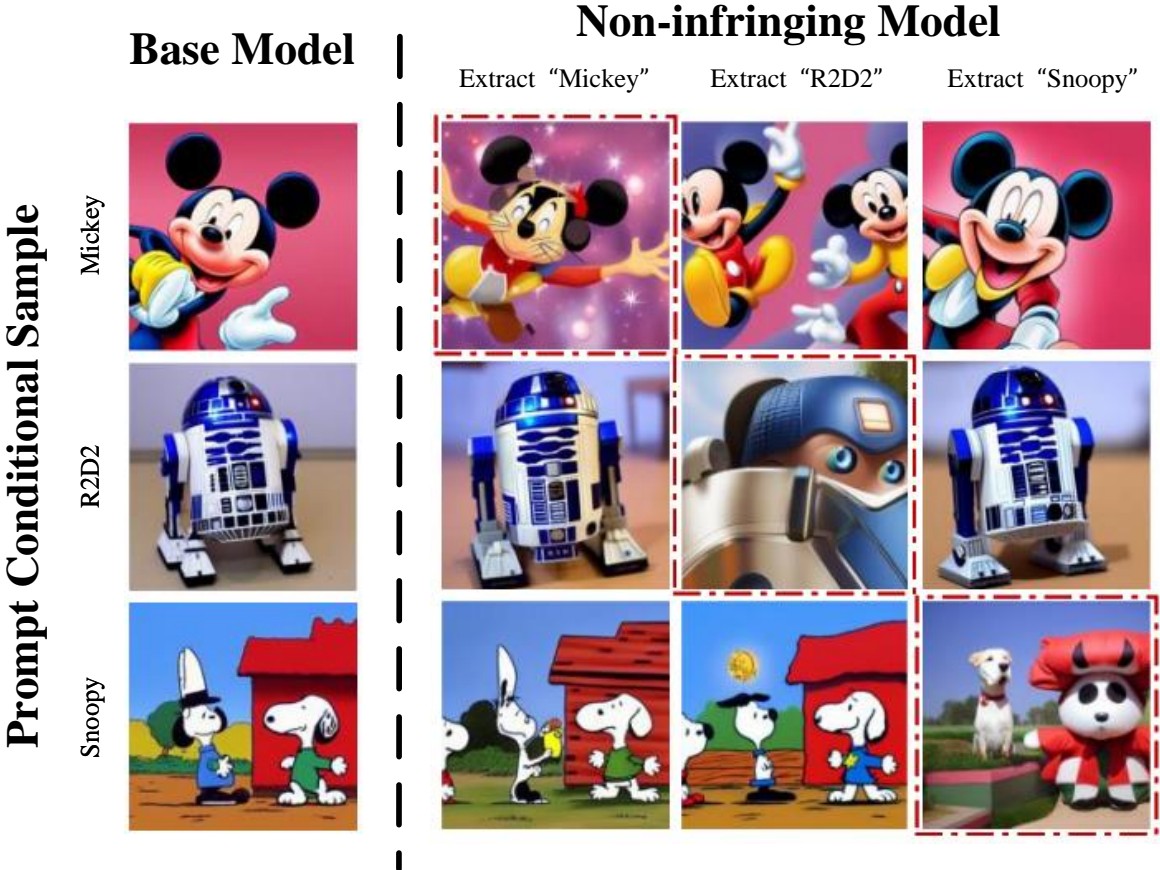

Figure 5: **Results of IP Recreation.** Each column on the right represents the output of a distinct non-infringing model. We successfully extract the unique IPs of Mickey, R2D2, and Vader independently, preserving the generation of other IPs.

Additionally, our comprehensive large-scale experiments demonstrate that the extraction process does not impact the model's ability to generate non-IP-related content. The performance of the model after extraction remains consistent with routine or everyday use cases, as further detailed in the appendix A.2. This ensures that while the model respects copyright constraints by effectively removing or isolating specific IPs, it does not compromise on its general utility or the breadth of its creative outputs.

Table 2: **Quantitative comparison in IP recreation.** We increase the KID of the target IP about 2.6 times compared with Concepts-Ablation while keeping the KID of the surrounding IP on par.

| Metrics | Methods | Target IP ↑ | Surrounding IP ↓ |
|---|---|---|---|
| KID$\times 10^3$ | EXTRACTION (OURS) | **131** | 17 |
| | CONCEPTS-ABLATION | 50 | 15 |

Furthermore, we demonstrate the combination of multiple IP ©plug-ins, as illustrated in Figure 6. The initial image is produced by the non-infringing model after extracting Mickey Mouse and Darth Vader, where the IPs are hard to recognize. Subsequent images, the second and third, are created after individually adding the Mickey and Vader ©plug-ins, which distinctly feature the respective IP. The final image is generated upon

adding the combined ©plug-in, successfully displaying both IPs. This procedure underscores the plug-in's efficacy in selectively and collectively restoring the model's ability to generate IP-specific content.

| **Prompt** | **Combined extraction** | **Mickey Integration** | **Vader Integration** | **Combinated Integration** |
|---|---|---|---|---|
| The cartoon of Mickey and Vader |  |  |  |  |

Figure 6: **IP Recreation in a single image.** We can integrate ©plug-in into the non-infringing model to generate either Mickey or Vader in a single image or integrate the combined ©plug-in to generate both of them.

These results indicate the efficacy of combining multiple extractions, where the non-infringing model's ability to produce images themed with either Mickey Mouse or Darth Vader is disabled. With the integration of the respective copyright plug-in, the model regains the capability to create content related to the specific IP.

## 4  Related Work

To position our work in the vast literature, we review related work through two perspectives: scope and technique. It is worth noting that some of the literature touches both sides and we organize them in a way most related to ours.

### 4.1  Scope Related: Copyright, Data Contribution and Credit Attribution

Recent text-to-image generative models are trained with large-scale datasets (Schuhmann et al., 2022; Liu et al., 2022), which cannot be guaranteed free of copyrighted data. At the same time, the state-of-the-art models are capable of generating high-quality and valuable creative images comparable to human creators or even memorizing the data points in the training set (Carlini et al., 2023b), which arouses copyright concerns about the training data and brings anxiety to the artist community.

Numerous efforts have been made for copyright protection of training data (Zhong et al., 2023). A direct approach is removing the copyrighted images from the training set, which may involve cumbersome costs due to the size of the training sets and may significantly degrade the model performance (Feldman, 2020). Another direct approach is post filtering, refusing to generate images with copyrighted concepts, e.g., Schramowski et al. (2023) proposes *Safe Latent Diffusion* to guide latent representation away from target concepts in the inference process, which nonetheless can be bypassed by a user with access to the model (Rando et al., 2022). As an example, OpenAI Dall·E3 (OpenAI, 2023) declines requests for generating an image in the style of a living artist and promises that creators can also opt their images out from training of future image generation models. Many papers discuss the idea of concept removal, which will be reviewed in later section.

Shan et al. (2023) propose *Image Cloaking* that suggests adding adversarial perturbations before posting artistic works on the internet so as to make them unlearnable for machine learning model, which has been pointed out to be hard to defend against future learning algorithms (Radiya-Dixit et al., 2021).

Theoretically, Bousquet et al. (2020); Elkin-Koren et al. (2023) connect the copyright protection of training data with the concept of differential privacy and discuss their subtle differences. Vyas et al. (2023) further formulate the copyright problem with a *near free access* (NAF) notion to bound the distance of the generative distributions of the models trained with and without the copyrighted data.

Our paper distinguishes largely from all previous works as we do not try to prohibit generating copyrighted concepts but instead we introduce a copyright authorization for the generative model to reward the copyright owners with fairness and transparency. From this aspect, our paper is also related with literature of monetizing the training data (Vincent & Hecht, 2021; Vincent et al., 2021; Li et al., 2022b;a) or attributing credits for the generative contents (Troynikov, 2023), but we establish a very distinct way to reward the authorship.

Heated discussion is also around the copyright for AI generated artwork Franceschelli & Musolesi (2022); Abbott & Rothman (2022). The Review Board of the United States Copyright Office has recently refused the copyright registration of a two-dimensional AI generated artwork entitled "A Recent Entrance to Paradise". However, Abbott & Rothman (2022) argues for giving the copyright to AI generated works, which will encourage people to develop and use creative AI, promote transparency and eventually benefit the public interest.

## 4.2 Technique Related: Concept Removal, and Negative Sampling

Our *extraction* operation is closely related with the *concept removal* for generative models. Rohit Gandikota and Joanna Materzyńska and Jaden Fiotto-Kaufman and David Bau (2023); Kumari et al. (2023) remove target concepts by matching the generation distribution of contexts with target concepts and that of contexts without target concepts. Zhang et al. (2023a) forget target concepts by minimizing the cross attention of target concepts with that of target images. Heng & Soh (2023) leverage the reverse process of continual learning to promote the controllable forgetting of target contents in deep generative models.

We note that negative sampling (Ho & Salimans, 2022) can also prevent generating certain concepts. Specifically, end users can set conditional context and negative context to guide the diffusion process to generate images conforming the conditional context while being far away from the negative context. Only negative sampling cannot stop copyright infringing generation because the contexts are set freely and adversarially by end users.

In contrast, for a specific copyrighted concept, our *extraction* operation takes an "Reverse LoRA" approach to disentangle the base model into two part: non-infringing base model and the plug-in LoRA component for copyrighted concept. Specifically, we use negative sampling to generate non-infringing images, which serves as training data for copyright plug-in. From the aspect of parameter efficient fine-tuning, our paper is related with literature (Alaluf et al., 2022; Ruiz et al., 2023; Gal et al., 2022; Hu et al., 2022; Huang et al., 2023).

Our *combination* operation is related with the widely studied "knowledge distillation" (Liang et al., 2023; Lopes et al., 2017; Sun et al., 2019; Hinton et al., 2015; Fang et al., 2019), but entails large difference from previous work. We combine multiple copyright plug-ins that are LoRA components for different targets, and we take data free approach due to practical constraint.

## 5 Discussion, Open Questions and Limitations

The growing concerns regarding generative AI models stem from their capacity to produce copyright-infringing content. This issue becomes more pronounced as state-of-the-art models continue to improve the quality of generated images, often without adequately acknowledging the contributions of human content creators. In response, we propose the "Copyright Plug-in Authorization" framework to address these societal worries, drawing inspiration from the purpose of copyright law. Our approach demonstrates that copyrighted data can be incorporated into LoRA plug-ins, enabling straightforward tracking of usage and equitable distribution of rewards.

A key challenge for this framework is the efficient management of a large number of plug-ins, which is essential to ensure user-friendly access to specific generations. Moreover, updating the base model poses another challenge, as retraining the entire suite of plug-ins can be costly, raising the issue of ensuring backward compatibility. One limitation of our current research is the potential degradation in the performance of the non-infringing model due to a large number of extraction operations, a factor that has yet to be thoroughly investigated.

## Acknowledgement

We would like to express our sincere gratitude to the anonymous reviewers for their constructive suggestions, which have significantly improved the quality of this manuscript. This work of Huishuai Zhang was supported (in part) by the State Key Laboratory of General Artificial Intelligence.

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

Copyright. `https://en.wikipedia.org/wiki/Copyright`.

AI Art Generators Spark Multiple Copyright Lawsuits. `https://www.hollywoodreporter.com`.

Ryan Abbott and Elizabeth Rothman. Disrupting creativity: Copyright law in the age of generative artificial intelligence. *Florida Law Review, Forthcoming*, 2022.

Yuval Alaluf, Omer Tov, Ron Mokady, Rinon Gal, and Amit Bermano. Hyperstyle: Stylegan inversion with hypernetworks for real image editing. In *Proceedings of the IEEE/CVF conference on computer Vision and pattern recognition*, pp. 18511–18521, 2022.

Mikołaj Bińkowski, Danica J Sutherland, Michael Arbel, and Arthur Gretton. Demystifying MMD GANs. In *International Conference on Learning Representations*, 2018.

Olivier Bousquet, Roi Livni, and Shay Moran. Synthetic data generators–sequential and private. *Advances in Neural Information Processing Systems*, 33:7114–7124, 2020.

Tom Brown, Benjamin Mann, Nick Ryder, Melanie Subbiah, Jared D Kaplan, Prafulla Dhariwal, Arvind Neelakantan, Pranav Shyam, Girish Sastry, Amanda Askell, et al. Language models are few-shot learners. In *Advances in Neural Information Processing Systems*, pp. 1877–1901. Curran Associates, Inc.

Nicholas Carlini, Florian Tramer, Eric Wallace, Matthew Jagielski, Ariel Herbert-Voss, Katherine Lee, Adam Roberts, Tom Brown, Dawn Song, Ulfar Erlingsson, et al. Extracting training data from large language models. In *30th USENIX Security Symposium (USENIX Security 21)*, pp. 2633–2650, 2021.

Nicholas Carlini, Daphne Ippolito, Matthew Jagielski, Katherine Lee, Florian Tramer, and Chiyuan Zhang. Quantifying memorization across neural language models. *International Conference on Learning Representations*, 2023a.

Nicolas Carlini, Jamie Hayes, Milad Nasr, Matthew Jagielski, Vikash Sehwag, Florian Tramer, Borja Balle, Daphne Ippolito, and Eric Wallace. Extracting training data from diffusion models. In *32nd USENIX Security Symposium (USENIX Security 23)*, pp. 5253–5270, 2023b.

Ali Edalati, Marzieh Tahaei, Ivan Kobyzev, Vahid Partovi Nia, James J Clark, and Mehdi Rezagholizadeh. Krona: Parameter efficient tuning with kronecker adapter. *arXiv preprint arXiv:2212.10650*, 2022.

Niva Elkin-Koren, Uri Hacohen, Roi Livni, and Shay Moran. Can copyright be reduced to privacy? *arXiv preprint arXiv:2305.14822*, 2023.

Gongfan Fang, Jie Song, Chengchao Shen, Xinchao Wang, Da Chen, and Mingli Song. Data-free adversarial distillation. *arXiv preprint arXiv:1912.11006*, 2019.

Vitaly Feldman. Does learning require memorization? a short tale about a long tail. In *Annual ACM SIGACT Symposium on Theory of Computing*, 2020.

Giorgio Franceschelli and Mirco Musolesi. Copyright in generative deep learning. *Data & Policy*, 4:e17, 2022.

Rinon Gal, Yuval Alaluf, Yuval Atzmon, Or Patashnik, Amit H Bermano, Gal Chechik, and Daniel Cohen-Or. An image is worth one word: Personalizing text-to-image generation using textual inversion. *arXiv preprint arXiv:2208.01618*, 2022.

Leo Gao, Stella Biderman, Sid Black, Laurence Golding, Travis Hoppe, Charles Foster, Jason Phang, Horace He, Anish Thite, Noa Nabeshima, et al. The Pile: An 800GB dataset of diverse text for language modeling. *arXiv preprint arXiv:2101.00027*, 2020.

Alvin Heng and Harold Soh. Selective amnesia: A continual learning approach to forgetting in deep generative models. *arXiv preprint arXiv:2305.10120*, 2023.

Martin Heusel, Hubert Ramsauer, Thomas Unterthiner, Bernhard Nessler, and Sepp Hochreiter. Gans trained by a two time-scale update rule converge to a local nash equilibrium. *Advances in neural information processing systems*, 30, 2017.

Geoffrey Hinton, Oriol Vinyals, and Jeff Dean. Distilling the knowledge in a neural network. *arXiv preprint arXiv:1503.02531*, 2015.

Jonathan Ho and Tim Salimans. Classifier-free diffusion guidance. *arXiv:2207.12598*, 2022.

Jonathan Ho, Ajay Jain, and Pieter Abbeel. Denoising diffusion probabilistic models. *Advances in neural information processing systems*, 33:6840–6851, 2020.

Edward J Hu, Yelong Shen, Phillip Wallis, Zeyuan Allen-Zhu, Yuanzhi Li, Shean Wang, Lu Wang, and Weizhu Chen. Lora: Low-rank adaptation of large language models. *International Conference on Learning Representations*, 2022.

Chengsong Huang, Qian Liu, Bill Yuchen Lin, Tianyu Pang, Chao Du, and Min Lin. Lorahub: Efficient cross-task generalization via dynamic lora composition. *arXiv preprint arXiv:2307.13269*, 2023.

Nam Hyeon-Woo, Moon Ye-Bin, and Tae-Hyun Oh. Fedpara: Low-rank hadamard product for communication-efficient federated learning. In *International Conference on Learning Representations*, 2021.

Nupur Kumari, Bingliang Zhang, Sheng-Yu Wang, Eli Shechtman, Richard Zhang, and Jun-Yan Zhu. Ablating concepts in text-to-image diffusion models. In *ICCV*, 2023.

Brian Lester, Rami Al-Rfou, and Noah Constant. The power of scale for parameter-efficient prompt tuning. In *Proceedings of the 2021 Conference on Empirical Methods in Natural Language Processing (EMNLP)*, EMNLP '21. Association for Computational Linguistics, 2021.

Hanlin Li, Brent Hecht, and Stevie Chancellor. All that's happening behind the scenes: Putting the spotlight on volunteer moderator labor in reddit. In *Proceedings of the International AAAI Conference on Web and Social Media*, volume 16, pp. 584–595, 2022a.

Hanlin Li, Brent Hecht, and Stevie Chancellor. Measuring the monetary value of online volunteer work. In *Proceedings of the International AAAI Conference on Web and Social Media*, volume 16, pp. 596–606, 2022b.

Xiang Lisa Li and Percy Liang. Prefix-tuning: Optimizing continuous prompts for generation. In *Proceedings of the 59th Annual Meeting of the Association for Computational Linguistics and the 11th International Joint Conference on Natural Language Processing (Volume 1: Long Papers)*, ACL-IJCNLP '21, pp. 4582–4597. Association for Computational Linguistics, 2021.

Chen Liang, Simiao Zuo, Qingru Zhang, Pengcheng He, Weizhu Chen, and Tuo Zhao. Less is more: Task-aware layer-wise distillation for language model compression. In *International Conference on Machine Learning*, pp. 20852–20867. PMLR, 2023.

Tsung-Yi Lin, Michael Maire, Serge Belongie, Lubomir Bourdev, Ross Girshick, James Hays, Pietro Perona, Deva Ramanan, C. Lawrence Zitnick, and Piotr Dollár. Microsoft coco: Common objects in context, 2015.

Yulong Liu, Guibo Zhu, Bin Zhu, Qi Song, Guojing Ge, Haoran Chen, GuanHui Qiao, Ru Peng, Lingxiang Wu, and Jinqiao Wang. Taisu: A 166m large-scale high-quality dataset for chinese vision-language pre-training. *Advances in Neural Information Processing Systems*, 35:16705–16717, 2022.

Raphael Gontijo Lopes, Stefano Fenu, and Thad Starner. Data-free knowledge distillation for deep neural networks. *arXiv preprint arXiv:1710.07535*, 2017.

OpenAI. Dall·E 3. `https://openai.com/dall-e-3`, 2023.

Maxime Oquab, Timothée Darcet, Théo Moutakanni, Huy Vo, Marc Szafraniec, Vasil Khalidov, Pierre Fernandez, Daniel Haziza, Francisco Massa, Alaaeldin El-Nouby, et al. Dinov2: Learning robust visual features without supervision. *arXiv preprint arXiv:2304.07193*, 2023.

Alec Radford, Jong Wook Kim, Chris Hallacy, Aditya Ramesh, Gabriel Goh, et al. Learning transferable visual models from natural language supervision. In *International conference on machine learning*, pp. 8748–8763. PMLR, 2021.

Evani Radiya-Dixit, Sanghyun Hong, Nicholas Carlini, and Florian Tramer. Data poisoning won't save you from facial recognition. In *International Conference on Learning Representations*, 2021.

Colin Raffel, Noam Shazeer, Adam Roberts, Katherine Lee, Sharan Narang, Michael Matena, Yanqi Zhou, Wei Li, and Peter J Liu. Exploring the limits of transfer learning with a unified text-to-text transformer. *The Journal of Machine Learning Research*, 21(1):5485–5551, 2020.

Aditya Ramesh, Mikhail Pavlov, Gabriel Goh, Scott Gray, Chelsea Voss, Alec Radford, Mark Chen, and Ilya Sutskever. Zero-shot text-to-image generation. In *International Conference on Machine Learning*, pp. 8821–8831. PMLR, 2021.

Aditya Ramesh, Prafulla Dhariwal, Alex Nichol, Casey Chu, and Mark Chen. Hierarchical text-conditional image generation with clip latents. *arXiv preprint arXiv:2204.06125*, 1(2):3, 2022.

Javier Rando, Daniel Paleka, David Lindner, Lennart Heim, and Florian Tramer. Red-teaming the stable diffusion safety filter. In *NeurIPS ML Safety Workshop*, 2022.

Rohit Gandikota and Joanna Materzyńska and Jaden Fiotto-Kaufman and David Bau. Erasing concepts from diffusion models. In *Proceedings of the 2023 IEEE International Conference on Computer Vision*, 2023.

Robin Rombach, Andreas Blattmann, Dominik Lorenz, Patrick Esser, and Björn Ommer. High-resolution image synthesis with latent diffusion models. In *Proceedings of the IEEE/CVF Conference on Computer Vision and Pattern Recognition (CVPR)*, pp. 10684–10695, June 2022a.

Robin Rombach, Andreas Blattmann, Dominik Lorenz, Patrick Esser, and Björn Ommer. High-resolution image synthesis with latent diffusion models. In *Proceedings of the IEEE/CVF conference on computer vision and pattern recognition*, pp. 10684–10695, 2022b.

Nataniel Ruiz, Yuanzhen Li, Varun Jampani, Yael Pritch, Michael Rubinstein, and Kfir Aberman. Dreambooth: Fine tuning text-to-image diffusion models for subject-driven generation. In *Proceedings of the IEEE/CVF Conference on Computer Vision and Pattern Recognition*, pp. 22500–22510, 2023.

Patrick Schramowski, Manuel Brack, Björn Deiseroth, and Kristian Kersting. Safe latent diffusion: Mitigating inappropriate degeneration in diffusion models. In *Proceedings of the IEEE Conference on Computer Vision and Pattern Recognition (CVPR)*, 2023.

Christoph Schuhmann, Romain Beaumont, Richard Vencu, Cade Gordon, Ross Wightman, Mehdi Cherti, Theo Coombes, Aarush Katta, Clayton Mullis, Mitchell Wortsman, et al. Laion-5b: An open large-scale dataset for training next generation image-text models. *Advances in Neural Information Processing Systems*, 35:25278–25294, 2022.

Shawn Shan, Jenna Cryan, Emily Wenger, Haitao Zheng, Rana Hanocka, and Ben Y. Zhao. Glaze: protecting artists from style mimicry by text-to-image models. In *Proceedings of the 32nd USENIX Conference on Security Symposium*, SEC '23, USA, 2023. USENIX Association. ISBN 978-1-939133-37-3.

Jascha Sohl-Dickstein, Eric Weiss, Niru Maheswaranathan, and Surya Ganguli. Deep unsupervised learning using nonequilibrium thermodynamics. In *International conference on machine learning*, pp. 2256–2265. PMLR, 2015.

Yang Song, Jascha Sohl-Dickstein, Diederik P Kingma, Abhishek Kumar, Stefano Ermon, and Ben Poole. Score-based generative modeling through stochastic differential equations. In *International Conference on Learning Representations*, 2020.

Siqi Sun, Yu Cheng, Zhe Gan, and Jingjing Liu. Patient knowledge distillation for bert model compression. In *Proceedings of the 2019 Conference on Empirical Methods in Natural Language Processing and the 9th International Joint Conference on Natural Language Processing (EMNLP-IJCNLP)*, pp. 4323–4332, 2019.

Hugo Touvron, Louis Martin, Kevin Stone, Peter Albert, Amjad Almahairi, Yasmine Babaei, Nikolay Bashlykov, Soumya Batra, Prajjwal Bhargava, Shruti Bhosale, et al. Llama 2: Open foundation and fine-tuned chat models. *arXiv preprint arXiv:2307.09288*, 2023.

Anton Troynikov. Stable Attribution. `https://www.stableattribution.com`, 2023.

Nicholas Vincent and Brent Hecht. A deeper investigation of the importance of wikipedia links to search engine results. *Proceedings of the ACM on Human-Computer Interaction*, 5(CSCW1):1–15, 2021.

Nicholas Vincent, Hanlin Li, Nicole Tilly, Stevie Chancellor, and Brent Hecht. Data leverage: A framework for empowering the public in its relationship with technology companies. In *Proceedings of the 2021 ACM Conference on Fairness, Accountability, and Transparency*, pp. 215–227, 2021.

Nikhil Vyas, Sham Kakade, and Boaz Barak. Provable copyright protection for generative models. *arXiv preprint arXiv:2302.10870*, 2023.

Eric Zhang, Kai Wang, Xingqian Xu, Zhangyang Wang, and Humphrey Shi. Forget-me-not: Learning to forget in text-to-image diffusion models. *arXiv preprint arXiv:2303.17591*, 2023a.

Lvmin Zhang, Anyi Rao, and Maneesh Agrawala. Adding conditional control to text-to-image diffusion models. In *IEEE International Conference on Computer Vision (ICCV)*, 2023b.

Richard Zhang, Phillip Isola, Alexei A Efros, Eli Shechtman, and Oliver Wang. The unreasonable effectiveness of deep features as a perceptual metric. In *CVPR*, 2018.

Haonan Zhong, Jiamin Chang, Ziyue Yang, Tingmin Wu, Pathum Chamikara Mahawaga Arachchige, Chehara Pathmabandu, and Minhui Xue. Copyright protection and accountability of generative ai: Attack, watermarking and attribution. In *Companion Proceedings of the ACM Web Conference 2023*, pp. 94–98, 2023.

# A   Appendix

## A.1   Intermediate results of extraction

We present an intricate analysis of the *extraction* process, elucidating distinct phases that delineate the evolutionary trajectory of the non-infringing model's performance. Following the de-concept step, the model experiences a transient phase marked by a temporary impairment in its ability to generate semantically meaningful images. Subsequently, the re-context step engenders a noticeable restoration, enhancing the model's proficiency in generating semantically rich images. Importantly, despite this recuperation, the model retains its inherent limitation – the incapacity to generate artwork in the distinctive style synonymous with Picasso. This observation underscores the success of the de-concept step, wherein the LoRA component effectively captures and removes the target concept, leading to the temporary impairment in the non-infringing model. The subsequent re-context step rectifies this performance decrease without reintroducing any information about the target concept. In essence, the *extraction* process successfully achieves the targeted concept extraction.

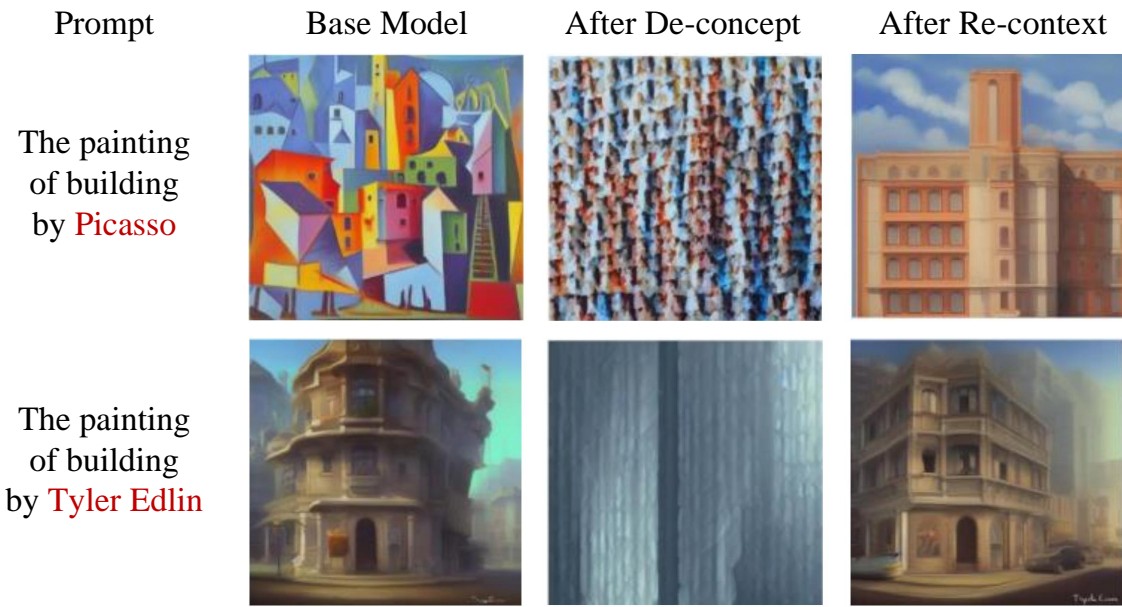

Figure 7: **Intermediate results of *extraction*.** After the de-concept step, the non-infringing model's generative abilities become significantly limited, predominantly manifesting as the production of noise. After the re-context step, the generation's prowess is rejuvenated, but due to the absence of learning Picasso-style images, the model remains unable to generate artwork in the style of Picasso.

## A.2   Experiment on ordinary objects generation

We evaluated the influence of *extraction* on the generation of ordinary objects. We utilize 5000 textual captions selected from the validation set in MS-COCO (Lin et al., 2015) as prompts, generating 5000 images using SD1.5 and the non-infringing model that extracts R2D2 and Picasso, respectively. Several randomly selected images are displayed in Figure 8. For illustrative purposes, we also generated results for concept-ablation and ESD on MS-COCO, respectively. Images within the same column exhibit substantial similarity, indicating that extraction does not exert an impact on the generation of ordinary items.

As depicted in Table3, we calculate some quantitative metrics like FID and KID. It is noteworthy that Concept-Ablation (Kumari et al., 2023) only releases the checkpoint of ablating "R2D2" and ESD (Rohit Gandikota and Joanna Materzyńska and Jaden Fiotto-Kaufman and David Bau, 2023) only releases the

Table 3: **Quantitative results on MS-COCO.** FID and KID metrics for removing the Picasso style are presented in the upper two rows, while those for removing R2D2 are displayed in the lower two rows.

| Domain | Method | FID ↓ | KID$\times 10^3$↓ |
|---|---|---|---|
| Style replication | *Extract Picasso* | 24.04 | 2.83 |
| | *Erase Picasso* | 25.20 | 3.39 |
| IP recreation | *Extract R2D2* | 20.55 | 2.36 |
| | *Ablate R2D2* | 18.97 | 1.34 |

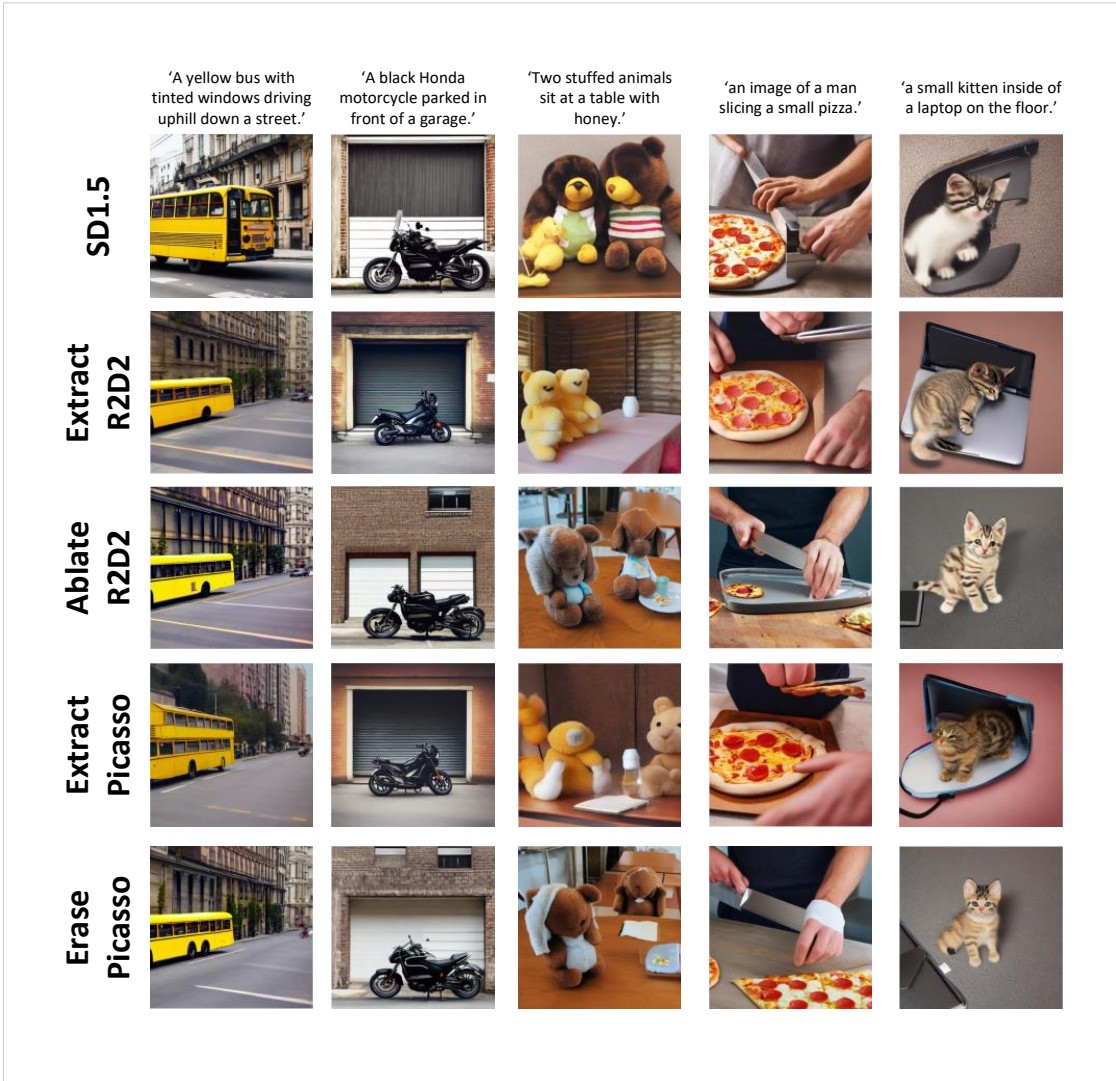

Figure 8: **Ordinary objects generation after *extraction*.** Row 1 displays images generated by Stable Diffusion V-1.5. Rows 2 and 3 illustrate images generated after the removal of the IP character R2D2, while Rows 4 and 5 showcase images generated after the elimination of Picasso's style. Rows 3 and 5 serve as the baseline, representing concept-ablation and ESD, respectively. Notably, after the *extraction* of R2D2 and Picasso, the non-infringing model retains the capability to generate commonplace objects sourced from the MS-COCO dataset (Lin et al., 2015).

checkpoint of erasing "Picasso". Thus, we compare them separately by using their respective checkpoints. Evaluation metrics consistently maintain low values, further affirming that extraction does not compromise the generation of ordinary items.

## A.3 More Quantitative Results

We compare the results of three methods following the removal of Van Gogh's influence. Like LPIPS, DINO-v2 Oquab et al. (2023) is a self-supervised model used for extracting visual features from images, allowing us to compute the similarity between generated images generated by the non-infringing model and the base model. The higher the value of DINO-v2, the closer the image is. Also, we calculate the CLIP-t distance to measure the semantic similarity between images generated by the non-infringing model and their corresponding textual prompts. The higher the CLIP-t value, the closer the sematic similarity is.

To ensure fairness in our evaluation, we implemented various settings to control the extraction effect and align it as closely as possible with the surrounding style. The quantitative results, as detailed in Table 4, demonstrate that our method achieves superior removal of the target style while preserving the integrity of the surrounding style.

Table 4: **Quantitative comparison of different methods.** The symbol ↑ indicates that the higher value is better on the metric, whereas ↓ symbol signifies the lower value is more preferable.The colors of symbols ↑ and ↓ should be read according to the colors of the most left column (Style).

| Style | Methods | KID$\times 10^3$↑↓ | LPIPS↑↓ | DINO-v2%↓↑ | CLIP-t%↓↑ |
|---|---|---|---|---|---|
| **Target Style** | ESD | 138 | 0.385 | 46.06 | 26.9 |
| | EXTRACTION (OURS) | 187 | 0.387 | 38.8 | 23.9 |
| **Surrounding Style** | ESD | 27 | 0.212 | 71.42 | 32.15 |
| | EXTRACTION (OURS) | 32 | 0.157 | 79.02 | 31.9 |
| **Target Style** | CONCEPT-ABLATION | 42 | 0.255 | 52.1 | 28.1 |
| | EXTRACTION (OURS) | 58 | 0.293 | 48.6 | 28.2 |
| **Surrounding Style** | CONCEPT-ABLATION | 12 | 0.123 | 81.2 | 29.6 |
| | EXTRACTION (OURS) | 7.4 | 0.115 | 86.2 | 32.8 |

## A.4 Experiment on seen and unseen contents generation

In *extraction*, we sample 10 common contents (training set) leveraging ChatGPT to fine-tune the base model. These contents have been previously processed by the non-infringing model. Additionally, we generate 10 supplementary contents for evaluation. Figure 9 shows the images generated with these 20 contents. The images on the left represent the seen contents (training set), while those on the right are the unseen contents (evaluation set). Within each image block, we extract the corresponding artistic style. The top row is generated by the base model, wheras the bottom row is generated by the non-infringing model. All images are generated with the prompt "[content] by [artist]". The specifics of the artistic and contents are detailed in Table 5.

To evaluate the performance of the base model and the non-infringing model, we computed various image distance metrics, including KID, LPIPS, and DINO-v2, between images generated by these two models. Additionally, we assessed the CLIP-t distance to measure the semantic alignment between textual prompts and images produced by the non-infringing model.

The quantitative results, averaged on the seen and unseen contents separately, are summarized in Table 6. The lack of significant differences across all metrics for the seen versus unseen content suggests that the model did not exhibit signs of over-fitting.

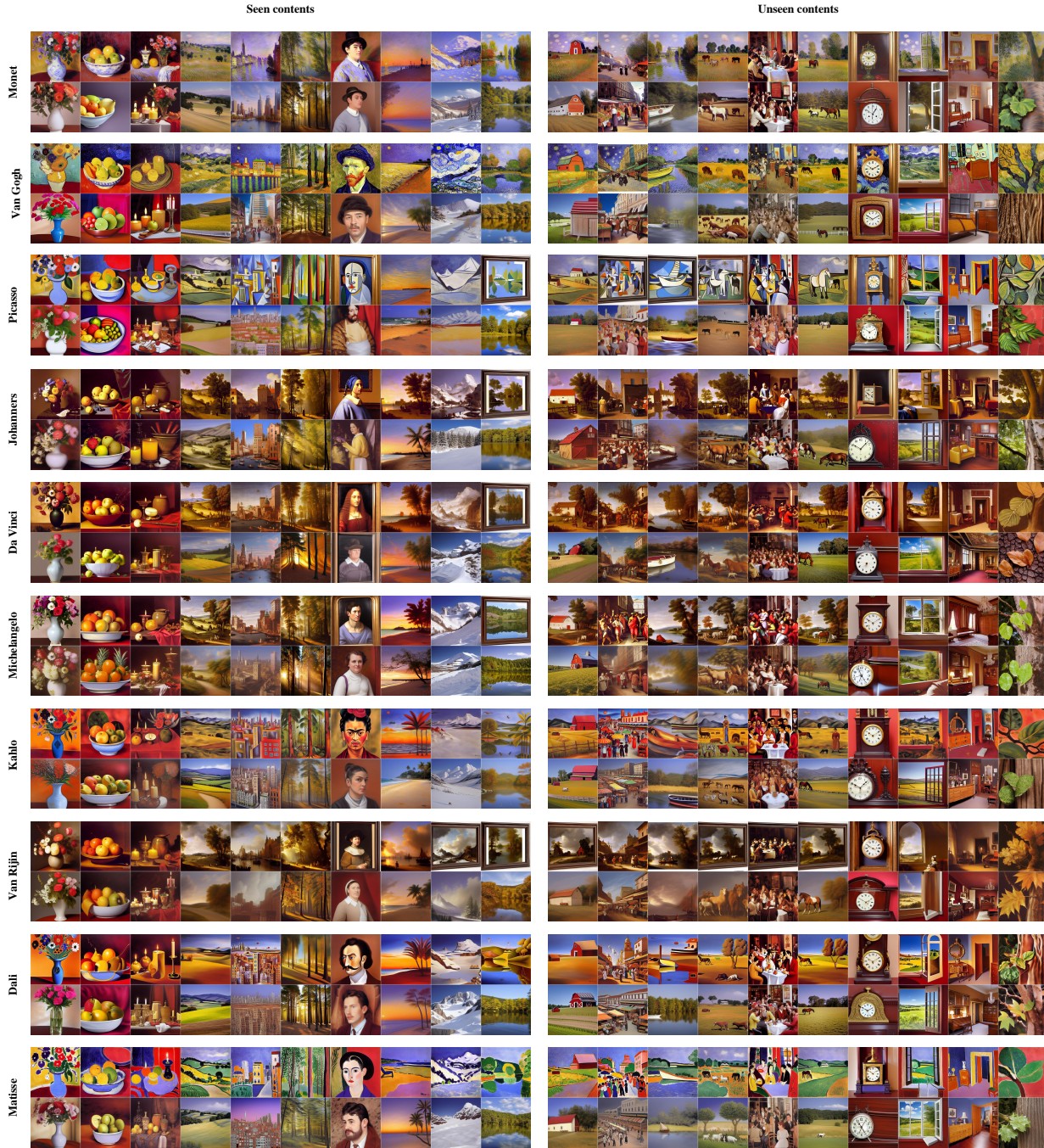

Figure 9: **Seen and Unseen contents generation after *extraction*.** On the left are 10 contents already seen during the *extraction*, while on the right aren't seen. Within each image block, the top row is generated by the base model, and the non-infringing model generates the bottom one. Zoom in for better visualization.

## A.5 Extracting specific prompt v.s. extracting style

Figure 10 compares the outputs of the base model and the non-infringing model under two different extraction scenarios, extracting specific prmpt "sunflowers by van gogh" and entire style "van gogh". For each scenario presented, the left column exhibits images generated by the base model, whereas the right column features images produced by the non-infringing model. The results indicate that the extraction method is effective

Table 5: **Details of artistic style and contents**

| Artistic Style | Seen Contents (training set) | Unseen Contents (evaluation set) |
|---|---|---|
| Claude Monet | vase of flowers | barn in a rural setting |
| Vincent van Gogh | bowl of fruit | bustling street market |
| Pablo Picasso | still life with candles | boat on a calm river |
| Johannes Vermeer | landscape with rolling hills | group of animals in a field |
| Leonardo da Vinci | cityscape with buildings | crowded café scene |
| Michelangelo | forest with sunlight filtering through trees | horse grazing in a pasture |
| Frida Kahlo | portrait of a person | vintage clock on a mantelpiece |
| Rembrandt van Rijn | quiet beach at sunset | window with a view of the countryside |
| Salvador Dalí | mountain range with snow | room with antique furniture |
| Henri Matisse | tranquil lake with reflections | close-up of a tree's bark and leaves |

Table 6: **Quantitative results on seen and unseen contents.** The gap in metrics between seen content and unseen content is not significant. The symbol ↑ indicates that the higher value is better on the metric, whereas ↓ symbol signifies the lower value is more preferable.

| Style | Contents | $\text{KID}\times10^3$↑↓ | LPIPS↑↓ | DINO-v2%↓↑ | CLIP-t%↓↑ |
|---|---|---|---|---|---|
| **Target Style** | *Seen Contents* | 137 | 0.315 | 46.8 | 28.5 |
| | *Unseen Contents* | 168 | 0.303 | 43.3 | 28.0 |
| **Surrounding Style** | *Seen Contents* | 17.92 | 0.139 | 81.0 | 32.3 |
| | *Unseen Contents* | 16.66 | 0.135 | 81.6 | 32.1 |

not only in isolating and modifying individual content-style associations but also in comprehensively altering an artist's entire style.

Table 7 shows that both extracting specific prompts and styles have little impact on the model's generative ability.

Table 7: **Quantitative results on COCO caption set.**

| Method | FID↓ | $\text{KID}\times10^3$ ↓ |
|---|---|---|
| Extract "van gogh" | 22 | 3.4 |
| Extract "sunflower by van gogh" | 20.8 | 1.4 |

### A.6 Style-IP ©plug-in combination

Figure 11 shows the combination of style and IP ©plug-ins. The initial image is generated by the non-infringing model after extracting R2D2 and van Gogh, where the IP and style ate hard to recognize. Subsequent images, the second and third, are produced by the model after separately adding the R2D2 and van Gogh ©plug-in, thus distinctly featuring the respective IP and style. The final image is created after adding the combined ©plug-in, successfully achieving IP recreation and style replication.

### A.7 The limitation of potential degradation in the performance of the non-infringing model

We observe a decline in the performance of the non-infringing model as the number of extracted styles increases. This degradation can be effectively mitigated by increasing the rate of re-context iterations and de-concept iterations, i.e., we can sample more contexts in the re-contexts in the re-context sub-process to maintain the generative ability of the non-infringing model.

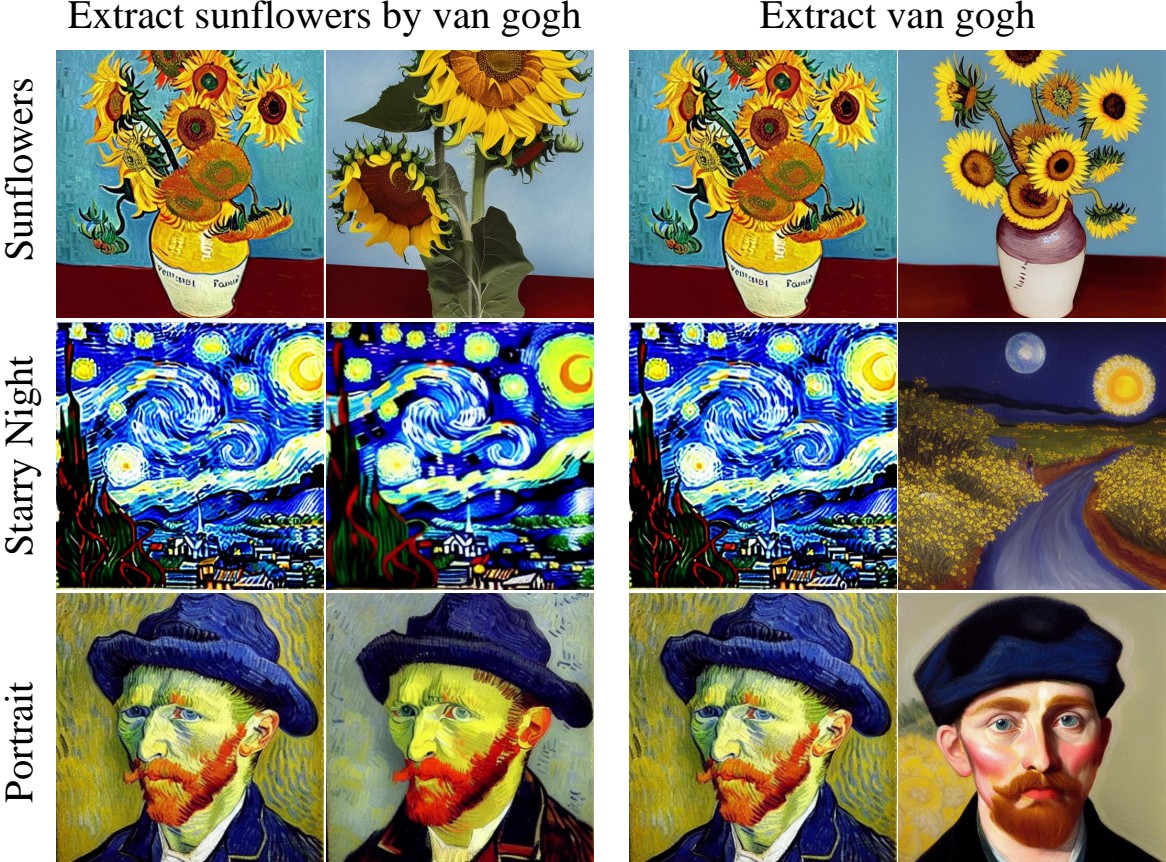

Figure 10: **The results of the non-infringing model after extracting specific prompt and style.** In the left block, only the specific prompt "sunflowers by van gogh" is extracted, while in the right block the entire style "van gogh" is extracted. For each block, the left column displays images generated by the base model, whereas the right column shows images generated by the corresponding non-infringing model.

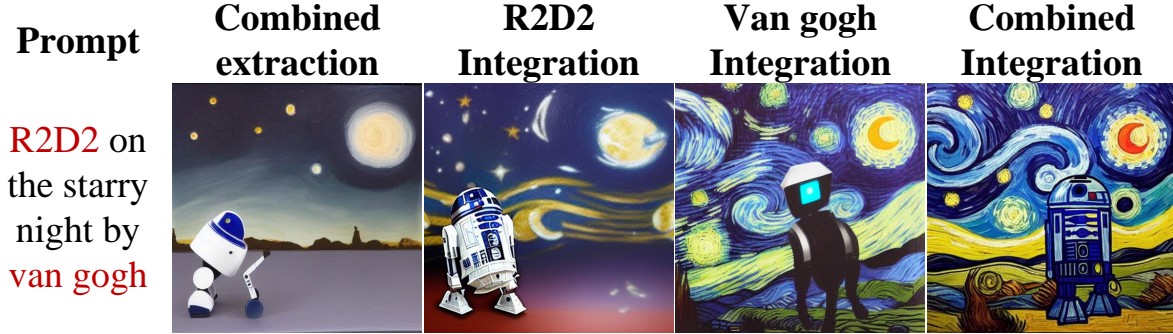

Figure 11: **IP Recreation and style replication in a single image.** We can integrate ©plug-in into the non-infringing model to recreate R2D2 or replicate van Gogh's style in a single image. Also, we can integrate the combined ©plug-in to achieve them both in an image.

During the *extract* process, we set the number of de-concept iterations to be 10 and set the number of re-context iterations to $10 \times r$. The default value for the rate is 1. For *extraction*, we leverage ChatGPT to generate $10 \times r$ common contents. For each iteration, we select one of these contents to generate 8 images. A higher $r$ allows the re-context phase to learn more contextual information, thereby maintaining the model's ability to generate high-quality images even as the number of extracted styles grows.

Figure 12 illustrates this phenomenon. The first row displays the image generated by the base model. Subsequent rows, from top to bottom, are images generated by non-infringing models that extract 1 to 10 styles, respectively. As the number of extracted styles increases, there is a noticeable decline in image quality. However, by increasing the rate, this degradation is significantly alleviated, as evidenced by improved image fidelity. The styles and contexts are detailed in Table 5. When the rate is greater than 1, we use the context in the evaluation set for the re-context phase.

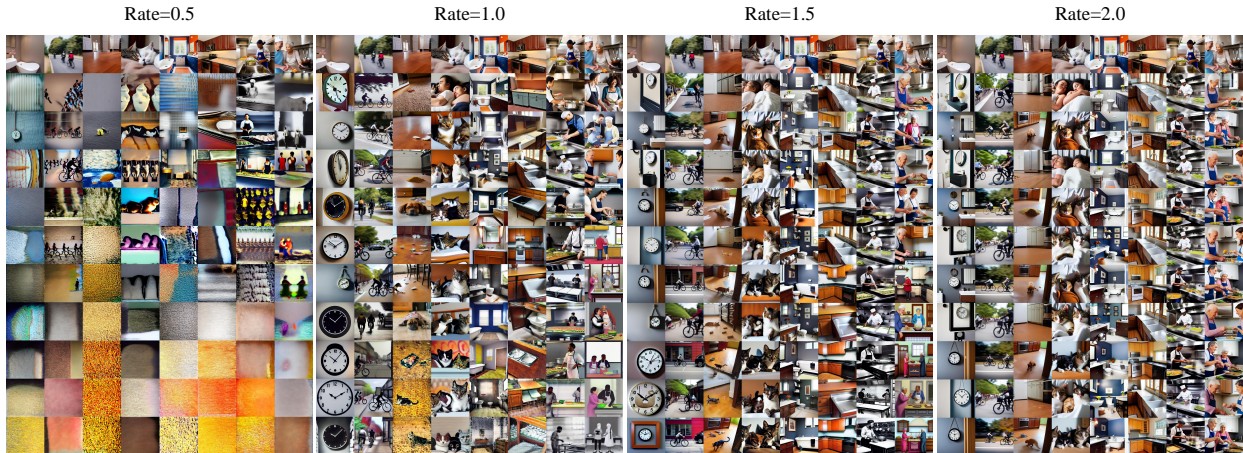

Figure 12: **The degradation and alleviation in the performance of the non-infringing model.** We randomly sample 8 images generated by different models. The first line is the image generated by the base model. The rest, from top to bottom, are images generated by non-infringing models that extract 1, 2, 3..., 10 styles, respectively. As the extracted concept number increases, the quality of images continuously declines. With the increase of the rate of re-context iteration number and de-concept iteration number, this decline has been effectively alleviated. Zoom in for better visualization.

