# OpenReview forum: "\copyright Plug-in Authorization for Human Copyright Protection in Text-to-Image Model"
_TMLR — Accepted by TMLR_

### Review · Reviewer_L9GJ · 2024-11-01

**Summary Of Contributions:**

The paper proposes a plug-in authorization framework that learns to disentangle a specific prompt from a pretrained image generation model into a lora module. This plug-in lora module can act as an authorization key to avoid copyright infringement. Only with this plug-in module can the base image generation model generate images related to the concept encoded in the lora module. The author also provided some discussions on methods to merge multiple concepts into a single lora module and included some examples to demonstrate the effect of the plug-in modules.

**Audience:**

Yes

**Claims And Evidence:**

No

**Requested Changes:**

See weaknesses

**Strengths And Weaknesses:**

Strengths:
1. The overall concept of a plug-in component in the image generation model that authorizes a certain IP is interesting and could be very useful for real-world applications of large-scale pretrained image generation models.
2. Experimental results demonstrate the effectiveness of the method for removing and restoring simple concepts.

Weaknesses:
1. The proposed plug-in lora component seems to be related to a specific prompt, instead of an IP or a whole concept. For example, I would like to know after extracting "Sunflower by Van Gogh" from a pretrained SD model, can the model still generate valid images based other Van Gogh-related prompts (e.g. Starry night by Van Gogh, etc)? If the model can still generate such images, then it's probably very hard for the method to scale up as essentially every prompt needs a different lora for plug-in authorization.
2. What is the $w-w_L$ operation mentioned in Section 2.2.1? Is it subtracting the Lora output from the pretrained SD output? Can the author provide more intuition on why this operation leads to the model forgetting the generation related to the IP specific prompt, but in the meantime doesn't affect the generation of other objects too much?
3. The experiments on the model's generation performance after extracting a certain prompt are very important and shouldn't be placed in the appendix. For the experiment in Appendix A.2 and Table 3, I'd also like to see the comparison of FID between pretrained Stable Diffusion model and the model after extracting a certain prompt. The method will not make sense if extracting a prompt from the model will dramatically hurt the image generation performance of the base model.

Minor concerns:
1. In Section 1, first paragraph: "these models not only excel at generating content based on user prompts cha..." Is this "cha" a typo or a broken reference to the citation?
2. In Section 2 first paragraph: ""As detailed in the Introduction..." There is a " symbol at the beginning of the paragraph.

---

> ### Author Response · Authors · 2024-12-27
> **Reply to Reviewer L9GJ**
>
> Thank you very much for your thorough review and valuable feedback on our manuscript. We are grateful for recognizing the potential real-world applications of our plug-in component and acknowledging the effectiveness of our method in removing and restoring simple concepts.
>
> For the weakness:
>
> 1. Plug-in seems to be related to a specific prompt instead of the whole concept:
>
> Thanks for asking this question.
> Our experiments verify that the reviewer's concern is not necessary. For each concept extraction, we sample 10 context prompts generate various images with the same concept to be extracted. These variations of prompts make sure the Plug-in captures the actual concept rather than specific prompt.
>
> Moreover, we add new experiments to verify the extraction process allow fine-grained control the "concept". We can extract only the "sunflower by Van Goah" style and do not affect other Van Gogh painting styles. We can also extract all the "Van Gogh" painting styles. Please see Figure 10 in the newly added Appendix A.7.
>
> Furthermore, Table1 shows that both extracting a specific paingting style and extracting the entire artist styles do not significantly degrade the image generation ability of the non-infringing model.
>
> Table 1: **Quantitative results on COCO caption set.** Both extracting a specific prompt and extracting the entire style won't hurt the image generation ability.
> | Method                           | FID↓ | KID$\times 10^3$↓ |
> |----------------------------------|------|------|
> | extract "van gogh"               | 22   | 3.4  |
> | extract "sunflower by van gogh"  | 20.8 | 1.4  |
>
> 2. Meaning of the $w-w_L$:
>
> It means that subtracting the LoRA weight from the pretrained Stable Diffusion model weight. Figure 7 in Appendix A.1 provides the intermediate results after De-concept ($w-w_L$), i.e., the generation capability of target concept has been extracted in the De-concept process. Then, the generation of surrounding objects is repaired in the Re-context process.
>
> 3. Typo problem:
>
> "Cha" is the reference to the citation. " " " is just a typo. We apologize for any confusion and have revised them.

---

### Review · Reviewer_LCJv · 2024-11-03

**Summary Of Contributions:**

This paper proposes a LoRA-based method to extract and combine copyright plug-ins for T2I diffusion models. This addresses the copyright infringement issues commonly encountered in the generative AI industry. The reported results demonstrate the effectiveness of the proposed method in extracting specific IPs and disentangling protected concepts from general concepts.

**Audience:**

Yes

**Claims And Evidence:**

Yes

**Requested Changes:**

Please refer to the weaknesses.

**Strengths And Weaknesses:**

**Strengths**

- The problem addressed is meaningful and worth studying.
- The proposed method is relatively complete, covering both extraction and combination.
- The results show the effectiveness of the proposed method.

**Weaknesses**

- Main concern: The authors use ChatGPT to extract 10 context prompts for a given artistic style or IP character during training and employ the same set of prompts for evaluation. I am concerned that the model may become overfitted to this specific prompt set and may not generalize to a broader variety of prompts. If this is the case, the copyright protection may ultimately be ineffective.
- While addition is mentioned in the paper, it is actually an original contribution of Civitai. The authors should highlight this earlier in the introduction to avoid misleading the audience.
- Why do the authors compare their method with concept ablation only on KID and with ESD only on LPIPS? It is recommended to compare both methods using both metrics.
- The results in the paper only include combinations of multiple artistic styles or multiple IP characters. Can the authors provide results that demonstrate the combination of artistic styles with IP characters?

---

> ### Author Response · Authors · 2024-12-27
> **Reply to Reviewer LCJv**
>
> Thank you very much for your insightful comments on our manuscript. We are grateful for recognizing the significance and completeness of our proposed method. We agree that addressing copyright concerns in generative models is a meaningful challenge, and we are pleased that our work contributes positively to this area.
>
> For the weakness:
>
> 1. Over-fitting Concern with ChatGPT Prompts:
>
> Thanks for proposing this question. We have done extra experiments and verified that such over-fitting effect is negligible. For evaluation, we resample 10 new contexts, distinct from those used during training. Quantitative results are presented in Table 1, showing that there is a negligible performance gap between the training set and the evaluation set. For better illustration, the visual results comparing the training set and the evaluation set are provided in Appendix A.4.
>
> Table 1:  **Quantitative results on seen and unseen contents.** The symbol $\uparrow$ indicates that the higher value is better on the metric, whereas the symbol $\downarrow$ indicates that the lower value is preferable. The left arrow represents "Target Style", while the right one is "Surrounding Style". The gap in metrics between seen content and unseen content is not significant.
> | Style               | Contents           | KID$×10^3$↑↓ | LPIPS↑↓ | DINO-v2%↓↑ | CLIP-t%↓↑ |
> |---------------------|--------------------|------------|---------|------------|-----------|
> | **Target Style**    | Seen Contents      | 137        | 0.315    | 46.8       | 28.5      |
> |                     | Unseen Contents    | 168        | 0.303    | 43.3       | 28.0      |
> | **Surrounding Style** | Seen Contents      | 17.9      | 0.139 | 81.0       | 32.3      |
> |                     | Unseen Contents    | 16.7      | 0.135    | 81.6       | 32.1      |
>
> 2. Clarification on "Addition".
>
> Thank you for your suggestion. We will revise the introduction to clearly attribute this "addition" contribution to existing projects, like DreamBooth and Civiti.
>
> 3. Comparison Metrics:
>
> In our initial submission,  we chose only two metrics KID and LPIPS for convenient comparison with existing works the ESD approach and the Concept-Ablation approach which use the LPIPS metric and the   KID   metric, respectively.
>
> During the rebuttal, we take the reviewer's suggestion and expand the evaluation metrics. As shown in Table 2, we now include additional metrics DINO-v2 and CLIP-t.
> For fair comparison, we design two settings, with each aligning the removal performance of target style with the existing two approaches, respectively. The results show that our method can achieve superior removal of the target style while maintaining the integrity of the surrounding style well.
> The details of this extended analysis are provided in Appendix A.3.
>
> Table 2: **Quantitative comparison of different methods.** The symbol $\uparrow$ indicates that the higher value is better on the metric, whereas the symbol $\downarrow$ indicates that the lower value is preferable. The left arrow represents "Target Style", while the right one is "Surrounding Style".
> | Style               | Methods                        | KID$×10^3$↑↓ | LPIPS↑↓ | DINO-v2%↓↑ | CLIP-t%↓↑ |
> |---------------------|--------------------------------|------------|---------|------------|-----------|
> | **Target Style**    | ESD                            | 138        | 0.385   | 46.06      | 26.9      |
> |                     | EXTRACTION (OURS)              | 187        | 0.387   | 38.8       | 23.9      |
> | **Surrounding Style** | ESD                            | 27         | 0.212   | 71.42      | 32.15     |
> |                     | EXTRACTION (OURS)              | 32         | 0.157   | 79.02      | 31.9      |
> | **Target Style**    | CONCEPT-ABLATION               | 42         | 0.255   | 52.1       | 28.1      |
> |                     | EXTRACTION (OURS)              | 58         | 0.293   | 48.6       | 28.2      |
> | **Surrounding Style** | CONCEPT-ABLATION               | 12         | 0.123   | 81.2       | 29.6      |
> |                     | EXTRACTION (OURS)              | 7.4        | 0.115   | 86.2       | 32.8      |
>
> 4. Combination of Artistic Styles with IP Characters:
>
> Thank you for asking this question. We have conducted experiments to combine artistic styles with IP characters, and the results are posted in Appendix A.6. We can remove both the style and the IP character, add either of them by integrating the corresponding plug-in, or add both of them to the non-infringing model by integrating the combined one at the same time.

---

> > ### Comment · Reviewer_LCJv · 2024-12-28
> > **Thanks**
> >
> > Thanks for the author's reply. My concerns are addressed.

---

### Review · Reviewer_t6UR · 2024-12-13

**Summary Of Contributions:**

The paper proposes a novel framework, Plug-in Authorization, to address copyright infringement concerns in text-to-image generative models. It introduces three operations—Addition, Extraction, and Combination—to handle copyright attribution and protection for artists and intellectual property (IP) owners. By leveraging methods like "Reverse LoRA" for extraction and "EasyMerge" for combination, the framework allows the separation of copyrighted styles or IPs from base models and the merging of multiple plug-ins into a cohesive model while maintaining non-infringing capabilities.

**Audience:**

Yes

**Broader Impact Concerns:**

N.A.

**Claims And Evidence:**

Yes

**Requested Changes:**

The requested changes are the same as the weaknesses.

**Strengths And Weaknesses:**

Strengths:

1. The paper proposes an innovative, scalable solution to copyright attribution in generative AI, directly aligning with existing intellectual property management practices.

2. From the technique perspective, Reverse LoRA is a method to disentangle copyrighted concepts from base models effectively. EasyMerge is a data-free, layer-wise distillation approach to combine multiple plug-ins seamlessly.


Weaknesses:

1. The evaluation metrics adopted in the work include KID and LPIPS metrics. More diverse performance metrics of T2I models could strengthen the argument.

2. The paper acknowledges potential performance degradation in non-infringing models after extensive extractions. It will be better to provide a quantitative or qualitative analysis of this impact.

---

> ### Author Response · Authors · 2024-12-27
> **Reply to Reviewer t6UR**
>
> Thank you for your constructive feedback on our manuscript. We are grateful for recognizing the innovation and scalability of our proposed framework, as well as the effectiveness of Reverse LoRA and EasyMerge methods.
>
> For the weakness:
>
> 1.  Evaluation Metrics:
>
> The KID and LPIPS metrics are respectively used in the Concept Ablation paper and the ESD paper. In our initial submission, we only test these two metrics for fair comparison with literature.
>
> As the reviewer suggested, we further evaluate two other model-based metrics DINO-v2 and CLIP-t and the results are presented in Table 1 below. It is known that there is always a trade-off between the removal of the target style and the retaining of the surrounding styles. Therefore, for a fair comparison, we compare the ESD method and the concept-ablation method separately because these two methods do not have the same level of removal performance. From Table 1, we can see that our extraction method achieves better or almost the same removal effect of target style as the existing methods while retaining better surrounding styles.  These new results are added in Appendix A.3 of the submission.
>
>
> Table 1: **Quantitative comparison of different methods**. The symbol $\uparrow$ indicates that the higher value is better on the metric, whereas $\downarrow$ symbol indicates that the lower value is preferable. The left arrow represents "Target Style", while the right one is "Surrounding Style".
> | Style               | Methods                        | KID×$10^3$↑↓ | LPIPS↑↓ | DINO-v2%↓↑ | CLIP-t%↓↑ |
> |---------------------|--------------------------------|------------|---------|------------|-----------|
> | **Target Style**    | ESD                            | 138        | 0.385   | 46.06      | 26.9      |
> |                     | EXTRACTION (OURS)              | 187        | 0.387   | 38.8       | 23.9      |
> | **Surrounding Style** | ESD                            | 27         | 0.212   | 71.42      | 32.15     |
> |                     | EXTRACTION (OURS)              | 32         | 0.157   | 79.02      | 31.9      |
> | **Target Style**    | CONCEPT-ABLATION               | 42         | 0.255   | 52.1       | 28.1      |
> |                     | EXTRACTION (OURS)              | 58         | 0.293   | 48.6       | 28.2      |
> | **Surrounding Style**| CONCEPT-ABLATION               | 12         | 0.123   | 81.2       | 29.6      |
> |                     | EXTRACTION (OURS)              | 7.4        | 0.115   | 86.2       | 32.8      |
>
> 2.  Performance Degradation:
>
> Thanks for proposing this question. Our extraction method admits sequential concept removals.
> However, as more concepts are to be removed, the error induced by each step of extraction could be accumulated and there could be performance degradation for the final non-infringing model as shown in the newly added Appendix A.7.
>
> There is an easy fix for this performance degradation, i.e., we can sample more contexts in the re-context sub-process to maintain the generative capability of the non-infringing model. We conduct preliminary experiments to show this. In the extraction process, we set the number of de-concept iterations to be $10$ and set the number of re-context iterations to be $10\times r$ with a default value $r=1.0$. For extraction, we use ChatGPT to generate $10\times r $ common contents.
> For each iteration, we select one of these contents to generate $8$ images.  A higher $r$ allows the re-context phase to learn more contextual information, thereby retaining the model's generation ability to create high-quality images even when the number of extracted concepts grows. These results are posted in the newly added Appendix A.7.

---

> > ### Comment · Reviewer_t6UR · 2024-12-28
> >
> > Thanks for the detailed reply! My concerns have been solved, and I lean on the acceptance of the paper.

---

### Decision · Action_Editor_1LoM · 2025-01-13

**Recommendation:** Accept as is

**Comment:**

The paper proposes an interesting and important problem, copyright protection, for generative models. It is of value to researchers in the area. However, the novelty is limited and hence, it doesn't meet the standards for certification or presentation at ICLR.

**Audience:**

Yes.

**Claims And Evidence:**

The claims are indeed supported by clear and convincing evidence. The problem studied -- plug-in authorization -- is an important one in the age of generative AI.